# Characterization of genetic diversity and population structure within *Staphylococcus chromogenes* by multilocus sequence typing

Rebeca Huebner[1], Robert Mugabi[2¤a], Gabriella Hetesy[2¤b], Lawrence Fox[3], Sarne De Vliegher[4], Anneleen De Visscher[4¤c], John W. Barlow[2]*, George Sensabaugh[1]

**1** Division of Infectious Diseases and Vaccinology, School of Public Health, University of California, Berkeley, California, United States of America, **2** Department of Animal and Veterinary Sciences, University of Vermont, Burlington, Vermont, United States of America, **3** College of Veterinary Medicine, Washington State University, Pullman, WA, United States of America, **4** M-team *and* Mastitis and Milk Quality Research Unit, Department of Reproduction, Obstetrics, and Herd Health, Faculty of Veterinary Medicine, Ghent University, Merelbeke, Belgium

¤a  Current address: Department of Veterinary Diagnostic and Production Animal Medicine, College of Veterinary Medicine, Iowa State University, Ames, Iowa, United States of America
¤b  Current address: Royal Veterinary College, Camden campus, London, United Kingdom
¤c  Current address: Flanders Research Institute for Agriculture, Fisheries and Food (ILVO), Technology and Food Science, Agricultural Engineering, Merelbeke, Belgium
* john.barlow@uvm.edu

**Data Availability Statement:** Data are available at the PubMLST website (https://pubmlst.org/ schromogenes). Others can access the data in the

## Abstract

*Staphylococcus chromogenes* is a common skin commensal in cattle and has been identified as a frequent cause of bovine mastitis and intramammary infections. We have developed a seven locus Multilocus Sequence Typing (MLST) scheme for typing *S. chromogenes*. Sequence-based typing systems, such as MLST, have application in studies of genetic diversity, population structure, and epidemiology, including studies of strain variation as a factor in pathogenicity or host adaptation. The *S. chromogenes* scheme was tested on 120 isolates collected from three geographic locations, Vermont and Washington State in the United States and Belgium. A total of 46 sequence types (STs) were identified with most of the STs being location specific. The utility of the typing scheme is indicated by a discrimination power of 95.6% for all isolates and greater than 90% for isolates from each of the three locations. Phylogenetic analysis placed 39 of the 46 STs into single core group consistent with a common genetic lineage; the STs in this group differ by less than 0.5% at the nucleotide sequence level. Most of the diversification in this lineage group can be attributed to mutation; recombination plays a limited role. This lineage group includes two clusters of single nucleotide variants in starburst configurations indicative of recent clonal expansion; nearly 50% of the isolates sampled in this study are in these two clusters. The remaining seven STs were set apart from the core group by having alleles with highly variable sequences at one or more loci. Recombination had a higher impact than mutation in the diversification of these outlier STs. Alleles with hypervariable sequences were detected at five of the seven loci used in the MLST scheme; the average sequence distances between the hypervariable alleles and the common core alleles ranged from 12 to 34 nucleotides.

same manner as the authors. Data are also in the Supporting Information files.

**Funding:** This study was supported by a University of Vermont College of Agriculture and Life Sciences competitive USDA HATCH experiment station multi-state project (NE1748) award (NEVT-H01920MS) and USDA Animal Health and Disease Research award (VT-AH02703) received by JB. The automated DNA sequencing was performed in Vermont Integrative Genomics Resource DNA Facility, which is supported by University of Vermont Cancer Center, Lake Champlain Cancer Research Organization, and the UVM Larner College of Medicine. The funders had no role in study design, data collection and analysis, decision to publish, or preparation of the manuscript.

**Competing interests:** The authors have declared that no competing interests exist.

The extent of these sequence differences suggests the hypervariable alleles may be remnants of an ancestral genotype.

## Introduction

*Staphylococcus chromogenes* was first recognized by Devriese et al. [1] as one of two subspecies of *Staphylococcus hyicus*, and was subsequently elevated to a novel species based on chemical, physiological and DNA-DNA re-association binding experiments [2]. Phylogenetic analyses by multi-locus and whole genome sequencing place *S. chromogenes* in a cluster with *S. hyicus* and *Staphylococcus agnetis* [3, 4]. The habitat of *S. chromogenes* is described as the body surface of cattle, pigs and poultry [2].

*S. chromogenes* is most commonly identified as a skin commensal and opportunistic mammary pathogen in cattle, sheep, goats and milking buffalo. It is a frequent cause of bovine mastitis [5], and reported as a skin pathogen of pigs and goats and as a cause of caprine mastitis [6–8]. *S. chromogenes* is recognized as one of the most frequent species of non-*aureus* staphylococci causing subclinical (asymptomatic) intramammary infections in dairy cattle in Europe and the United States [reviewed in 5 and 9]. *S. chromogenes* has been identified as a cause of persistent intramammary infections in dairy cattle [10–12], and infections appear to be associated with increased milk somatic cell counts (i.e. intramammary inflammation or subclinical mastitis) [12–14]. The organism has also been identified from extra-mammary skin swabs of cattle, including udder skin, teat apex, and streak canal [15–18]; compared to some other non-*aureus Staphylococcus* species, *S. chromogenes* is less commonly isolated from environmental sites in surveys of dairy farm environmental sources (e.g. barn air, surfaces and bedding) [19]. Some authors have suggested that *S. chromogenes* intramammary infection or colonization of teat skin may have a protective effect against *S. aureus* mastitis [20, 21]. Using PFGE, multiple strains (pulsotypes) of *S. chromogenes* have been isolated from intramammary infections and extramammary skin sites of dairy cattle within individual herds [15, 18]. *S. chromogenes* has been isolated infrequently from nasal swabs of humans in close contact with cattle [9].

Pulsed field gel electrophoresis (PFGE), ribotyping, amplification fragment length polymorphism (AFLP), random amplification of polymorphic DNA (RAPD)-PCR, and multiple locus variable-number tandem repeat analysis (MLVA) have been used to discriminate between *S. chromogenes* strains in epidemiological studies [10, 11, 15, 18, 19, 22, 23]; these methods demonstrate sufficient discriminatory power in short-term epidemiological investigations. However, DNA fragment electrophoretic band pattern (i.e. "DNA fragment fingerprinting") methods are limited in portability and reproducibility between laboratories and by "an inability to quantitate genetic relationships between isolates" [24]. Sequence-based typing systems, such as multilocus sequence typing (MLST), have been recommended to improve our understanding of the epidemiology of *S. chromogenes* [5]; sequence-based typing also offers the potential to characterize phylogenetic relationships and population dynamics from a local to a global scale [24].

In this paper we report the development of a MLST scheme that provides a practical, portable, sequence-based approach for the identification of strain types and the characterization of relationships between clonal lineages in *S. chromogenes*. MLST schemes have been developed for a number of staphylococcal species including *S. aureus*, *S. epidermidis*, *S. haemolyticus*, *S. hominis*, *S. lugdunensis*, *S. pseudintermedius*, and *S. carnosus* [25–31]. This *S. chromogenes* MLST scheme is based on the detection of genetic variation in seven housekeeping genes in

120 isolates collected from dairy cattle (*Bos taurus*) in three geographic locations, Vermont and Washington State in the United States and Belgium. This sample population allows assessment of both genetic and geographic diversity present in this species. The scheme has been designed such that the seven loci are well separated around the ca. 2.34 Mb genome of *S. chromogenes* to maximize the opportunity to evaluate the extent to which recombination may play a role in shaping diversity at the population level.

## Material and methods

### Bacterial strains and DNA isolation

A convenience sample of 120 isolates, from collections of three laboratories, was investigated in this study; these isolates originated from dairy cattle in Vermont (n = 46) and Washington (n = 24) in the USA and from Belgium (n = 48), and pigs in Vermont (n = 2), (S1 Table). The isolates from Belgium were collected in 2001 and 2012 from multiple farms from dairy cattle teat apex swabs (n = 20) and individual mammary quarter milk samples from apparent healthy quarters (n = 28) [14, 17, 19]. The Washington isolates were collected in 2010 from quarter milk samples of dairy cows with intramammary infections on 6 farms in Washington and Idaho [32]. The Vermont isolates were collected in 1998 from 3 dairy farms and in 2013 from 2 dairy farms from either quarter milk samples of cows with intramammary infections (n = 31), cow teat orifice swabs (n = 1), cow hock skin swabs (n = 3), and bulk tank milk (n = 11); two isolates originated from pig nasal swabs were collected in 2013 from one of the Vermont farms. This study was carried out in strict accordance with the recommendations in the Guide for the Care and Use of Laboratory Animals of the National Institutes of Health. The protocol was approved by the Committee on the Ethics of Animal Experiments of the University of Vermont (Protocol Number: 13–033).

Isolates from Washington were previously identified as *S. chromogenes* using partial *16S rRNA* and *rpoB* gene sequencing, *gap* gene PCR-restriction fragment length polymorphism and API STAPH ID 20 biochemical testing (bioMérieux, Inc. Durham, NC) [32]. Isolates from Belgium were previously identified as *S. chromogenes* using transfer RNA intergenic spacer PCR or *16S rRNA* gene fragment sequencing [14, 17]. Isolates collected in Vermont in 1998 were previously identified as *S. chromogenes* using API STAPH ID 20 biochemical testing. Subsequently all 120 isolates were verified as *S. chromogenes* by sequence analysis of *tuf* and *rpoB* gene amplicon fragments with > 97% sequence identity [33, 34]. The draft genome sequence of *S. chromogenes* strain MU970 was downloaded from the NCBI microbial genome database (GenBank accession JMJF01000000) to provide a genome reference sequence [12]. All isolates were shared between the University of California Berkeley (UCB) and University of Vermont (UVM) laboratories, and DNA extraction, amplification and analysis procedures were replicated in both labs. Strain MU970 (gift from J. Middleton, University of Missouri) was cultured for DNA extraction and sequence amplification in the UVM lab.

Each laboratory used a different pre-existing protocol for primary growth and preparation of isolates for DNA extraction. In the University of California Berkeley (UCB) lab, isolates were grown aerobically overnight in tryptic soy broth (TSB, BBL) at 37˚C without shaking and then plated on tryptic soy agar plates with 5% sheep blood (TSA, BBL) and incubated aerobically at 37˚C. Single colonies were passed from each TSA plate and grown overnight at 37˚C in 1 mL of TSB. To achieve mid-phase growth, 100μL of overnight growth was combined with 5mL TSB and incubated at 37˚C with shaking for 4 hours. Cell pellets were collected from 3 mL of mid-phase growth by centrifugation at 10,000 rpm for 5 minutes. Alternatively, in parallel, at the University of Vermont lab (UVM), a pure primary culture was grown aerobically for 48hrs on TSA, and single colonies from this growth were inoculated to 5 ml TSB, grown

aerobically overnight at 37˚C and cell pellets were collected by centrifugation directly from 1.8 ml of overnight TSB culture. The cell pellets were then frozen at -20˚C until DNA extraction could be performed (UCB) or held at 4˚C and processed within 48 hours (UVM).

DNA extraction was performed using a DNeasy Blood & Tissue kit (Qiagen, Valencia, CA, USA) according to manufacturer's instructions with the modification that the initial lysis buffer was supplemented with lysostaphin (22 U/ml; Sigma-Aldrich). DNA yield and quality was assessed by electrophoresis using a 0.75% agarose gel containing the DNA stain GelStar (Lonza, Rockland, ME, USA) in 1X Tris Borate EDTA (TBE) buffer. Aliquots of DNA were stored at -20˚C.

## Selection of target loci for MLST

Fifteen loci were assessed initially as potential candidates for the *S. chromogenes* MLST scheme. Nine loci originated from MLST schemes used for *S. aureus*, *S. epidermidis*, and *S. saprophyticus* [http://pubmlst.org and unpublished] and six additional loci were selected from the Gen-Bank annotation listing for the *S. chromogenes* MU970 draft genome. PCR primers for each of the candidate loci were designed using Primer-BLAST to yield sequence segments 700–850 bp in length. The candidate loci were evaluated for sequence variation using a panel of 27 isolates from Vermont and those exhibiting the greatest sequence diversity were selected for further evaluation. As a final filter, we sought to assess whether the loci were relatively evenly distributed around the genome to maximize the potential for diversity resulting from inter-locus recombination. This was determined initially by locating the positions of the 7 candidate loci on the complete genome sequence of the closely related species, *S. hyicus* [GenBank accession CP008747.1]; subsequent localization on the recently reported complete genome sequence of *S. chromogenes* strain 1401 [NZ_CP04602.1] established a minimum inter-locus distance of 240Kb with an average distance of 335Kb. Details for the seven loci selected for the MLST scheme are provided in S2 Table.

## Target gene amplification and nucleotide sequencing

Target genes were amplified using the polymerase chain reaction employing a master mix containing 16.75μL DNase-free H2O, 2.5μL PCR buffer, 1.5μL 50mM MgCl2, 0.5μL 10mM dNTP, and 0.25μL 0.5 U/μL Taq polymerase (Invitrogen) per reaction. Master mix was aliquoted into tubes for each locus being amplified and 0.25μL each of 25μM forward and reverse primer was added for each reaction. Three microliters of DNA was added to 22 μL of the master mix and primers. PCR cycling included heating to 95˚C for 7 minutes, followed by 35 cycles of 94˚C for 45 seconds, 55˚C for 45 seconds and 2 minutes of 72˚C. On the last cycle, the samples were heated to 72˚C for 5 min. The samples were then held at 4˚C until further analysis could be completed.

Amplification products (3μL) were evaluated by electrophoresis at 150V for approximately one hour on 1.5% agarose gel containing the DNA stain GelStar. PCR products exhibiting a single band at the predicted amplicon size were processed for sequence analysis by the UC Berkeley sequencing facility. PCR amplification and target gene sequencing were replicated at UVM and the replicate sets of amplicons were processed for sequence analysis by the University of Vermont DNA sequencing facility.

## Analysis of MLST sequence data

Raw sequence data containing both forward and reverse reads were recorded in FASTA format for analysis. Sequences were aligned using the MUSCLE function in MEGA 6.06 [35]. After alignment, single strand overhangs and any ambiguous reads were trimmed from the ends of

each sequence. Any missing or ambiguous nucleotides were resolved by reviewing the trace data using the FinchTV 1.4.0. viewer. Once a consensus sequence was determined for each sample, all sequences for a single locus were combined into a single FASTA file. The sequences were trimmed again to obtain a standard length. Sites exhibiting single nucleotide polymorphism (SNP) were identified in MEGA. For each of the seven loci, the gene sequence present in the MU970 reference sequence was arbitrarily defined as allele 1 and new alleles were identified by pairwise comparison of SNP sites; each new allele was assigned a number. Sequence types (STs) were defined by unique allelic profiles at the seven loci. MEGA and DnaSP 5.10 were used for assessment of population genetics parameters such as nucleotide diversity and allele (haplotype) diversity [35, 36]; the haplotype diversity is a measure of the discrimination power of the typing system [37]. DnaSP was also used to concatenate the sequences of the seven loci for each ST.

The 7-locus concatenated nucleotide sequence data were used for the construction of phylogenetic trees generated by the neighbor joining (NJ) algorithm in MEGA with 1000 bootstrap replications. Clonal clusters of sequence types were identified at the level of single locus variants (SLVs) and double locus variants (DLVs) using the web program eBurst3 as implemented in goeBURST [38, 39]. Evidence for recombination was assessed by surveying allelic sequences at each locus within clonal subgroups delineated by eBURST and phylogenetic analyses: alleles at a locus within a clonal subgroup differing at a single nucleotide site were scored as mutations whereas alleles differing at multiple nucleotide sites and alleles shared between different clonal subgroups were scored as recombination events [40]. Recombination between loci was assessed using the four-gamete test [41] as implemented in DnaSP; this test detects the minimum number of recombination events (RM) in the history of the sample. For this test, the concatenated sequences were constructed with the first locus (*arcC*) sequences appended to the end of the 7-locus concatenation to detect possible recombination between the last and first locus in the circular genome. The pairwise homoplasy index (PHI) for recombination was measured using the program implemented in SplitsTree [42, 43]; recombination within and between sequences was considered positive if the PHI test yielded a $p \leq 0.05$.

## Results and discussion

### Genetic diversity in *S. chromogenes*

The MLST scheme for *S. chromogenes* is based on characterization of nucleotide sequence variation in fragments of seven housekeeping genes in 120 *S. chromogenes* isolates. Overall, 216 nucleotide substitutions at 213 sites were identified in the 4563 bp of genome sequence covered by the scheme (Table 1). The 216 nucleotide substitutions resulted in 57 amino acid replacements, a replacement rate of 26.4%. The number of alleles detected at the seven loci ranged from 9 to 21; the majority of alleles differ in amino acid sequence as well as nucleotide sequence. The average number of nucleotide differences per site at each locus in the sample population of 120 isolates is indicated by the nucleotide diversity ($\pi_p$). The allelic diversity (Hd) reflects the probability that any pair of isolates drawn from the sample population will carry different alleles at a locus; it is a measure of the discrimination power of the locus in the typing system [37]. The *arcC* locus exhibits the greatest nucleotide diversity followed by *dnaJ* and *glpF* among the 7 MLST loci; however, *glpF* is superior to *arcC* and *dnaJ* with regard to discrimination power. Sequences of the alleles at the each of the 7 loci are available at the PubMLST database (https://pubmlst.org/schromogenes).

A total of 46 distinct Sequence Types (STs) were identified in the sample population; the 7-locus allelic profiles of the 46 STs are listed along with their geographic origins in Table 2 and at the PubMLST database (https://pubmlst.org/schromogenes). By convention, the allele

**Table 1. Characterization of allelic sequence variation observed in 120 unique isolates of *S. chromogenes*.**

| Locus | Gene | Sequence length (bp) | No. Alleles | S ($\eta$) | Amino Acid Substitutions | Isolates (n = 120) | | STs (n = 46) | |
|---|---|---|---|---|---|---|---|---|---|
| | | | | | | $\pi_P$ | Hd | $\pi_s$ | Hd |
| *arcC* | Carbamate kinase | 588 | 21 | 70 (72) | 19 | 0.01545 | 0.690 | 0.01825 | 0.805 |
| *hutU* | Urocanate hydrase | 693 | 9 | 17 | 6 | 0.00195 | 0.330 | 0.00251 | 0.388 |
| *fumC* | Fumerate hydratase | 636 | 14 | 16 | 4 | 0.00207 | 0.569 | 0.00250 | 0.698 |
| *dnaJ* | chaperone protein dnaJ | 747 | 18 | 48 | 11 | 0.00678 | 0.613 | 0.00828 | 0.760 |
| *glpF* | glycerol uptake facilitator | 612 | 17 | 30 (31) | 8 | 0.00626 | 0.836 | 0.00737 | 0.871 |
| *menF* | Isochorismate synthase | 597 | 11 | 11 | 5 | 0.00095 | 0.341 | 0.00140 | 0.456 |
| *pta* | Phosphate acetyl transferase | 690 | 10 | 21 | 4 | 0.00275 | 0.591 | 0.00382 | 0.654 |
| | 7 Locus Averages | 652 | 14.3 | 30.4 | 8.1 | 0.00517 | 0.567 | 0.00630 | 0.662 |
| | 7 Locus Totals | 4563 | 46 | 213 (216) | 57 | | 0.956 | | |

S ($\eta$): number of polymorphic sites (number of mutations when different from number of polymorphic sites)

$\pi_P$: nucleotide diversity per site in the population of 120 isolates

$\pi_s$: nucleotide diversity per site in 46 STs

Hd: Allelic Diversity.

sequences in the reference strain MU970 were defined as allele 1 with the corresponding 7-locus allelic profile of ST1 for MU970. The average nucleotide diversity for the 46 STs is 0.00507; this is somewhat lower than the values 0.0068, 0.0064, and 0.010 for *S. aureus*, *S. epidermidis*, and *S. hominis* respectively [28] but higher than the 0.0021 value for *S. carnosus* [31] and much higher than the 0.00035 value for *S. haemolyticus* [27].

ST1 was the most common sequence type observed in the sample population; it was detected in isolates from all three source locations though primarily (17 out of 18) from the two US locales. Only three other STs were found in multiple locales: ST6 and ST18 in Vermont and Belgium and ST15 in both US locales. The remaining 42 STs were detected in only one of the three locales (Table 2). ST1 plus three additional STs (ST28, ST15, & ST5) account for over 1/3 (n = 44) of the isolates in the sample population. At the other end of the frequency spectrum, 24 STs were found only as single isolates. The remaining 52 isolates are distributed among 18 STs containing 2–4 isolates each. This spectrum of isolate distribution among STs is commonly observed in MLST characterization of other staphylococcal species: typically a relatively small number of STs contain a large proportion of the total isolates whereas many STs, often over half, are represented by single isolates [27–31].

The discrimination power of the 7-locus MLST scheme for strain characterization within the overall population is 95.6% (Table 1). Calculation of discrimination powers based only on the isolates present in each of the individual geographic populations ranged from 90.2% for the Vermont cohort to 93% for the Belgian cohort. This indicates that each of the three sample populations is genetically diverse despite apparent nearly complete genetic isolation from each other. These values are in the range typically seen with MLST schemes for staphylococcal species.

Previous studies have implicated *S. chromogenes* as a frequent cause of intramammary infections in cattle on dairy farms and have demonstrated strain diversity can vary within and between farms. However the DNA fragment profiling methods used in these studies to characterize strain diversity, notably PFGE, ALPF, RAPD-PCR, and MLVA, have yielded conflicting estimates of diversity, ranging from a high degree of genetic conservation to considerable genetic heterogeneity [10, 11, 18, 19, 22, 23, 44–49]. Given the stand-alone nature of these studies and the incompatibility of their measures of strain diversity, the results of different studies cannot be directly compared or consolidated. In contrast, the intrinsic portability of

**Table 2. MLST profiles of 46 STs and isolate origins.**

| ST | arcC | hutU | fumC | dnaJ | glpF | isoC | pta | N* | Vermont | Wash. | Belgium |
|---|---|---|---|---|---|---|---|---|---|---|---|
| ST1 | 1 | 1 | 1 | 1 | 1 | 1 | 1 | 18 | 11 | 6 | 1 |
| ST2 | 1 | 1 | 1 | 1 | 1 | 1 | 3 | 1 | 1 | | |
| ST3 | 1 | 1 | 1 | 1 | 1 | 1 | 6 | 1 | 1 | | |
| ST4 | 1 | 1 | 1 | 1 | 1 | 1 | 8 | 1 | | 1 | |
| ST5 | 1 | 1 | 1 | 1 | 4 | 1 | 1 | 7 | 7 | | |
| ST6 | 1 | 1 | 1 | 3 | 2 | 1 | 2 | 3 | 1 | | 2 |
| ST7 | 1 | 1 | 1 | 5 | 4 | 1 | 1 | 1 | 1 | | |
| ST8 | 1 | 1 | 1 | 8 | 1 | 1 | 1 | 1 | | 1 | |
| ST9 | 1 | 1 | 1 | 11 | 1 | 1 | 1 | 1 | | 1 | |
| ST10 | 1 | 1 | 5 | 3 | 2 | 1 | 2 | 4 | 4 | | |
| ST11 | 1 | 2 | 1 | 1 | 1 | 1 | 1 | 4 | 4 | | |
| ST12 | 1 | 6 | 1 | 1 | 1 | 1 | 3 | 1 | | 1 | |
| ST13 | 2 | 3 | 3 | 2 | 1 | 1 | 2 | 4 | 4 | | |
| ST14 | 3 | 1 | 1 | 1 | 3 | 3 | 2 | 1 | | 1 | |
| ST15 | 3 | 1 | 1 | 1 | 3 | 5 | 2 | 8 | 6 | 2 | |
| ST16 | 3 | 1 | 1 | 1 | 3 | 6 | 2 | 1 | | 1 | |
| ST17 | 3 | 1 | 1 | 1 | 10 | 1 | 2 | 4 | | 4 | |
| ST18 | 4 | 4 | 4 | 4 | 5 | 4 | 4 | 2 | 1 | | 1 |
| ST19 | 5 | 1 | 1 | 1 | 6 | 5 | 2 | 2 | 2 | | |
| ST20 | 6 | 1 | 6 | 2 | 7 | 1 | 2 | 2 | 2 | | |
| ST21 | 7 | 1 | 6 | 1 | 3 | 1 | 5 | 1 | 1 | | |
| ST22 | 8 | 5 | 14 | 6 | 3 | 8 | 2 | 1 | 1 | | |
| ST23 | 9 | 1 | 8 | 7 | 8 | 7 | 7 | 1 | 1 | | |
| ST24 | 10 | 1 | 6 | 1 | 9 | 1 | 5 | 1 | | 1 | |
| ST25 | 11 | 7 | 2 | 9 | 11 | 2 | 4 | 1 | | 1 | |
| ST26 | 12 | 8 | 7 | 10 | 12 | 1 | 2 | 3 | | 3 | |
| ST27 | 13 | 1 | 1 | 1 | 1 | 1 | 1 | 1 | | 1 | |
| ST28 | 1 | 1 | 1 | 1 | 2 | 1 | 2 | 11 | | | 11 |
| ST29 | 1 | 1 | 1 | 1 | 2 | 9 | 2 | 1 | | | 1 |
| ST30 | 1 | 1 | 1 | 3 | 15 | 1 | 2 | 4 | | | 4 |
| ST31 | 1 | 1 | 1 | 3 | 17 | 1 | 2 | 1 | | | 1 |
| ST32 | 1 | 1 | 6 | 14 | 3 | 1 | 2 | 2 | | | 2 |
| ST33 | 1 | 1 | 6 | 14 | 13 | 1 | 2 | 1 | | | 1 |
| ST34 | 1 | 1 | 9 | 1 | 1 | 1 | 1 | 1 | | | 1 |
| ST35 | 1 | 1 | 10 | 15 | 3 | 13 | 2 | 1 | | | 1 |
| ST36 | 3 | 1 | 1 | 12 | 10 | 1 | 2 | 1 | | | 1 |
| ST37 | 8 | 8 | 7 | 13 | 3 | 1 | 2 | 2 | | | 2 |
| ST38 | 14 | 1 | 6 | 1 | 3 | 1 | 2 | 4 | | | 4 |
| ST39 | 15 | 1 | 13 | 16 | 3 | 1 | 2 | 3 | | | 3 |
| ST40 | 16 | 1 | 6 | 1 | 2 | 1 | 2 | 1 | | | 1 |
| ST41 | 16 | 8 | 17 | 1 | 3 | 12 | 2 | 1 | | | 1 |
| ST42 | 17 | 1 | 1 | 18 | 2 | 1 | 2 | 2 | | | 2 |
| ST43 | 19 | 9 | 11 | 17 | 14 | 2 | 10 | 3 | | | 3 |
| ST44 | 20 | 1 | 6 | 1 | 5 | 1 | 5 | 2 | | | 2 |
| ST45 | 21 | 1 | 1 | 3 | 2 | 1 | 2 | 1 | | | 1 |
| ST46 | 23 | 1 | 6 | 3 | 16 | 1 | 9 | 2 | | | 2 |

*The number of isolates (N) detected for each ST and their geographic origins are indicated in the right hand columns

MLST sequence data coupled with the high discrimination power of this MLST system provide a uniform approach for wide-ranging investigations of strain diversity within and between locations or over time and for testing associations between strain type and isolate sources, disease states, and host species. Moreover, isolates represented in the ever expanding whole genome sequence databases can be characterized using MLST sequence data extracted *in silico* from the databases, thus allowing inter-compatibility of datasets from different studies [50].

It is to be noted that the seven gene loci employed in the MLST scheme for *S. chromogenes* are also present in the three related species: *S. hyicus*, *S. agnetis*, and *S. felis* [3]. A preliminary survey of the genomes of these three species indicate that all seven loci are genetically polymorphic, thus providing the potential for development of MLST schemes based on the same seven loci for each of the three species. With a sequence identity of less than 90% between the genes in the different species, the possibility of confusing the different MLST schemes is effectively eliminated.

## Population structure and geographic origins

Characterization of population structure using the eBURST algorithm groups STs according to the number of allele differences at the 7 loci; this approach disregards the extent of sequence difference between alleles. Initial analysis at the single locus variant (SLV) level revealed two clonal clusters, one centered on ST1 with 11 satellite STs and the other centered on ST6 with 7 satellite STs; in addition, there were several ST pairs and triplets. The ST6 cluster included ST28, the second most common ST in the population with nearly four times as many isolates than ST6, prompting the question of whether ST28 might be the founder of the cluster. Investigation at double locus variant (DLV) level showed 33 STs connected in a single network with ST28 at the central node with radiations leading to four secondary nodes centering on ST1, ST6, ST15, and ST38 (Fig 1). The 33 STs in this core network account for 96 of the 120 isolates in the sample population. The 13 STs not included in this network are separated from the network and from each other by sequence differences at 3 or more loci.

The ST1 and ST6 nodes connect to ST28 directly, ST1 as DLV and ST6 as a SLV. In terms of nucleotide distances, ST28 and ST1 differ at 4 SNP sites (3 in *glpF* and 1 in *pta*); ST28 and ST6 differ at 7 SNP sites in *dnaJ* allele 3. The *glpF* and *dnaJ* allele differences are more likely the result of recombination events given that the sequence motifs of both alleles are present in STs within and outside the common network. The ST15 and ST38 nodes, in contrast, connect to ST28 via intermediary DLV STs: ST17 and ST40 respectively. Despite this, the two nodes are relatively close to ST28 in nucleotide distance, differing at 6 SNP sites for ST15 and 7 SNP sites for ST38, again likely involving recombination events.

The cluster around ST1 consists entirely of single locus variants (SLVs), each bearing a different single nucleotide substitution. This starburst pattern is indicative of a recent clonal expansion with ST1 as the founder. Of the 11 STs in the ST1 cluster, all but one, ST34, originate from Vermont or Washington farms. The cluster around ST 15 is also primarily associated with isolates from Vermont and Washington farms. Unlike the ST1 cluster, the ST15 cluster consists mostly of DLVs. Despite this, the average nucleotide distance between ST15 and its satellites is 2.2.

The STs in the ST6 cluster and the ST38 cluster are predominantly of Belgian origin. The ST6 cluster consists of SLVs in which all but one of the linkages involves loci differing at a single SNP site. Again, this starburst pattern is indicative of a recent clonal expansion with ST6 as the founder. The cluster around ST38 contains more DLVs than SLVs. One ST in the ST38 cluster, ST44, separates itself from the other STs in the cluster by differing from them at an average of 58 SNP sites.

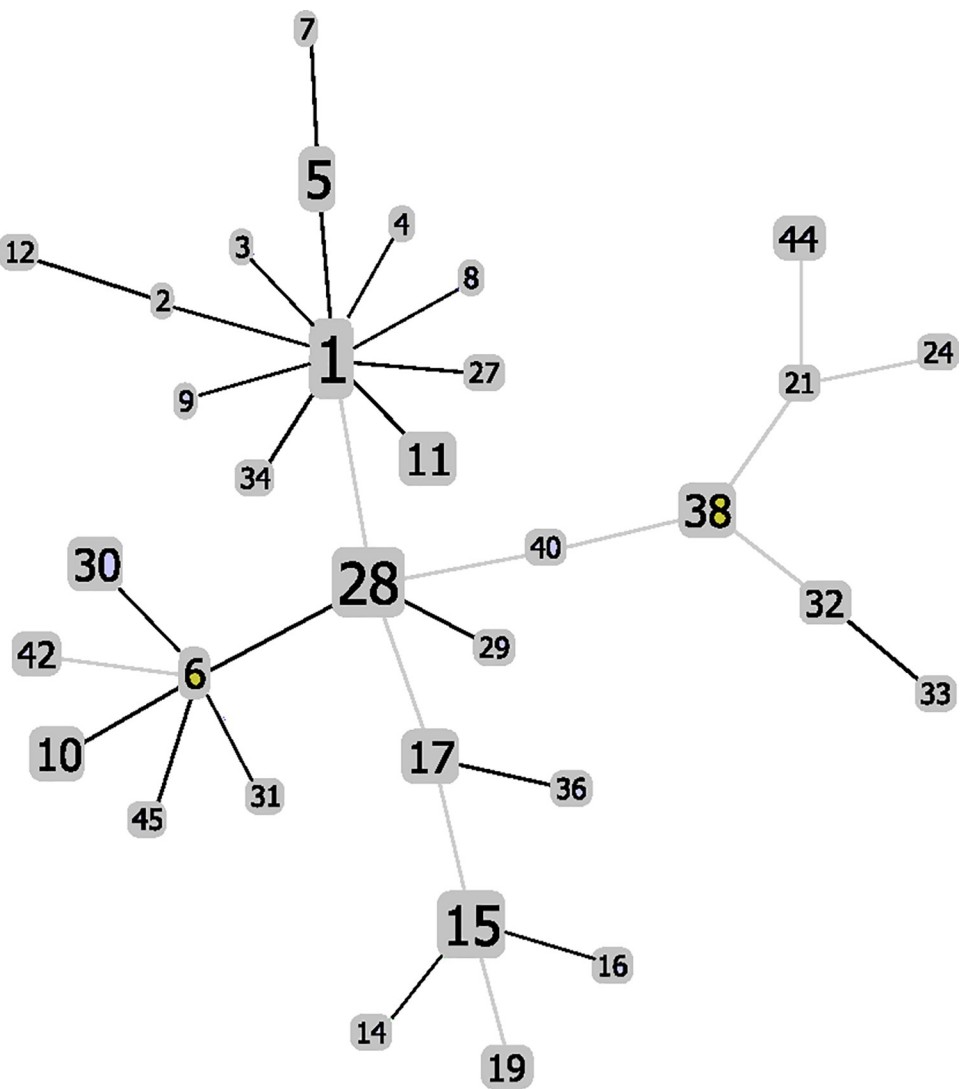

**Fig 1. Population structure of *S. chromogenes* as indicated by eBURST at the double locus variant (DLV) level.** Each of the 33 STs in the eBURST network is represented by a box, the size of which corresponds to the number of isolates in the ST. Heavy (black) lines represent single locus variants, light (grey) lines represent double locus variants.

The geographic partitioning of isolates within the core ST network suggests that *S. chromogenes* populations are relatively isolated, more so between Belgium and the United States than between Vermont and Washington within the U.S. Two lines of evidence suggest Europe as the historical origin of the strains of *S. chromogenes* investigated in this study: (a) Belgium is home to more STs in the core network than either of the U.S. source locations, and (b) the central node of the core network, ST28, is of Belgian origin. This hypothesis can be tested by characterizing MLST databases representing more geographically diverse sample populations should they be available in the future.

## Phylogenetic analysis distinguishes core and outlier STs

Phylogenetic analysis based on overall nucleotide sequence variation between the 46 STs provides an alternative perspective on the population structure of *S. chromogenes* (Fig 2). As

a

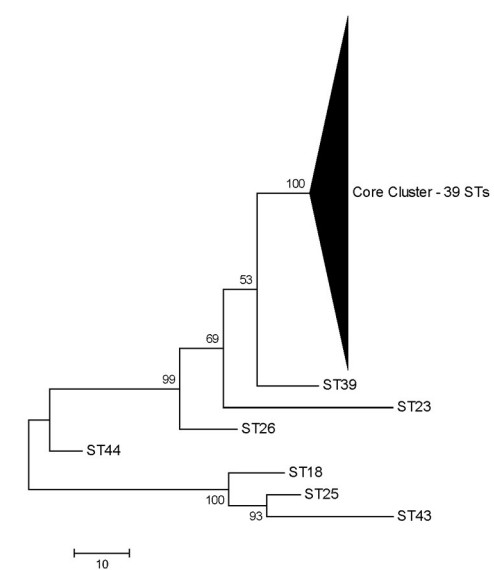

b

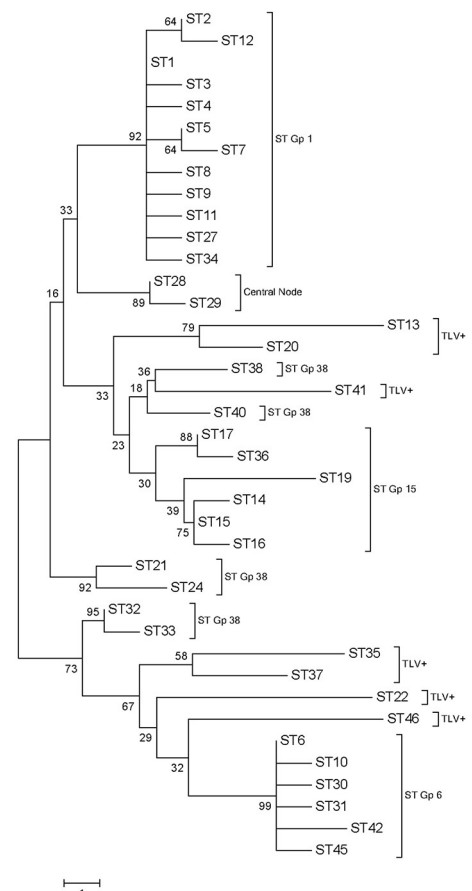

**Fig 2. Phylogenetic tree of the 46 STs of *S. chromogenes*.** The trees were constructed using the concatenated sequences of the seven MLST loci; bootstrap values are indicated at the branch points and the scale bar is in units of nucleotide differences. Fig 2A characterizes phylogenic relationships of all 46 STs. Fig 2B elaborates the relationships among the 39 STs in the large undifferentiated group in Fig 2A. Nodal clusters identified in the eBURST analysis are specified as ST groups, e.g., ST Gp1, ST Gp6, etc. STs differing at 3 or more loci from the eBURST groups are identified TLV+.

shown in Fig 2A, this analysis clusters 39 of the 46 STs into one large group with 100% bootstrap support. The remaining 7 STs are placed on separate branches with deeper roots.

The large cluster (Fig 2B) includes 32 of the 33 STs in the eBURST core network plus 7 STs differing at 3 or more loci from those in the eBURST core group (STs 13, 20, 22, 35, 37, 41, & 46). The one member of the eBURST core network that placed outside the phylogenetically defined large group was ST44, previously noted as differing substantially at the sequence level from the other STs in the eBURST network. Within the large cluster, only the STs in the ST Gp1 and ST Gp6 appear as unified clusters with strong bootstrap support; the remaining STs, including the 7 STs noted above, are interspersed on variably supported branches. The mean pairwise nucleotide distance between the 39 STs in the group is 9.6 (range 1–22), a relatively small increase over the mean distance of 8.1 (range 1–16) between the 32 STs in the eBurst core network. This increase in nucleotide distance is accounted for by the additional sequence variation present in the STs varying at three or more loci compared to the STs in the eBURST network which are single or double locus variants. The conjoining of the 32 STs in the eBURST network with the seven additional STs in the phylogenetic analysis is thus consistent with all 39 STs sharing a common genetic lineage that has undergone diversification. This grouping includes 105 of the 120 isolates in the total population set.

The placement of the remaining seven STs as outliers to the common core cluster does not reflect meaningful phylogenetic relationships. Rather it is the consequence of these seven STs carrying hypervariable allelic variants at one or more of the MLST loci. Pairwise comparisons of allele sequences at each of the 7 MLST loci show alleles at five loci partition into two classes, one consisting of alleles typically differing at 4 or fewer nucleotides and a second smaller group differing by 10 or more nucleotides from the first group. The alleles in the first group comprise the allelic composition of the 39 STs in the common core cluster and are designated here as common core alleles; alleles in the second group are designated hypervariable (HV). Table 3 compares the relationship of the two classes of alleles in terms of the average pairwise nucleotide distances within and between the classes. It is clear the distances between the two classes are substantially greater than the within-class distances. Two loci, *hutU* and *menF*, lack HV alleles.

**Table 3. Comparison of common core alleles and hypervariable (HV) alleles.**

| Locus | Common Core Alleles | | | Hypervariable alleles | | | Average Pairwise Distance. (nt) | | |
|---|---|---|---|---|---|---|---|---|---|
| | No. Alleles | SNP sites | a.a. subs. | SNP Sites | a.a. subs | Core | HV | Core vs. HV | |
| *arcC* | 14 | 14 | 11 | 58 | 8 | 3.2 | 20.3 | 34.5 |
| *hutU* | 6 | 5 | 3 | 3 | 2 | 1.7 | 2.0 | 12.2 |
| *fumC* | 14 | 16 | 4 | – | – | 3.6 | – | – |
| *dnaJ* | 14 | 15 | 7 | 32 | 4 | 4.9 | 17.0 | 20.0 |
| *glpF* | 14 | 16 | 8 | 21 | 2 | 3.3 | 2.7 | 17.8 |
| *menF* | 11 | 11 | 5 | – | – | 2.7 | – | – |
| *pta* | 7 | 6 | 4 | 6 | 0 | 1.9 | 4.0 | 13.1 |

Alleles at each locus were partitioned into common core and hypervariable groups; each group was characterized independently.

To illustrate the effect of a single HV allele in a MLST profile, STs 26 & 39 have HV alleles only at the *arcC* locus and are clear outliers in the 7-locus phylogeny (Fig 2A) but a phylogeny built on the six loci excluding *arcC* results in a repositioning of these two STs within the common core cluster. The outliers ST44 and ST23 differ from the common core with HV alleles at two and three loci respectively. The remaining three outlier STs (STs 18, 25, & 43) have HV alleles at the five loci and fall into a well-supported group with average nucleotide distances of 105.5 to 108.6 separating these three from the 39 STs in the common core cluster. Notably, these three STs also differ significantly from each other with an average pairwise nucleotide difference of 30.7 between them. These three STs represent 6 isolates of which 4 originate from Belgium and one each from Vermont and Washington State.

## Evidence of recombination in *S. chromogenes*

The contribution of mutation and recombination to the generation of genetic diversity varies considerably among staphylococcal species. In *S. aureus*, for example, new alleles are predominately generated by point mutations whereas in contrast, recombination and mutation contribute almost equally to the generation of new alleles in *S. haemolyticus* and the ratio is about 2:1 favoring recombination in *S. epidermidis* [26, 27, 51]. Species that undergo very low rates of recombination have population structures characterized by clonal lineages that diversify slowly by the accumulation of point mutations. At the other end of the spectrum, species that undergo frequent recombination can exhibit a level of genetic diversity that complicates phylogenetic analysis and reconstruction of population structure.

The pairwise homoplasy index (PHI) was used to gain an initial assessment of recombination among the concatenated sequences of the 32 STs in the eBurst network, the 39 STs in the common core, and the 46 STs in the full data set. No statistically significant evidence of recombination was detected for the core 32 STs (p = 0.80), but recombination was indicated for the 39 STs in the common core ($p$ = 0.018) and very strong evidence for recombination was found for the full set of 46 STs ($p < 0.0001$). To characterize the distribution of recombination events within and between loci, the "four gametes" test of Hudson and Kaplan [41] was used; this test yields the minimum number of recombination events between SNP positions in the concatenated ST sequences. Detection of recombination between MLST loci is of particular interest for it indicates expansion of genomic diversity beyond that provided by allele sequence variation. For the 32 ST sequences in the eBURST core complex, four inter-locus and no intra-locus recombination events were detected; the inter-locus recombinants were *arcC*/*fumC*, *fumC*/*dnaJ*, *dnaJ*/*glpF*, and *glpF*/*menF*. Analysis of the 39 ST sequences in the common core group added one more inter-locus recombination event, *menF*/*arcC*, plus an intra-locus recombination event in *dnaJ*. Analysis of all 46 ST sequences added 10 more within-locus events: 6 in *arcC*, 2 in *fumC*, and 2 in *glpF* for a total of 16 minimum recombination events overall. Analysis of the outlier 7 ST sequences accounted for 11 of these events, the five between loci and six within loci. These findings are consistent in showing that recombination contributes to genetic diversification in *S. chromogenes*, particularly in the STs with HV alleles.

To assess the relative contributions of mutation and recombination events at the allele level, allelic sequence changes were surveyed at each locus within the nodal subgroups delineated by eBURST. Alleles differing at a single nucleotide site were scored as mutations whereas alleles differing at multiple nucleotide sites and alleles shared between different clonal subgroups were scored as recombination events [40]. Notably, the defining allelic signature of three of the four nodal subgroups can be attributed to recombination events contributing one or more new alleles to the allelic profile of ST28, the central node. The nodes of the ST1 and ST6 nodal subgroups differ from ST28 by recombined alleles at the *glpF* and *dnaJ* loci respectively. In

contrast, the allelic differences between the STs within each nodal cluster are single nucleotide substitutions in keeping with the starburst topologies of these two nodal clusters. The ST15 nodal subgroup differs from ST28 with recombinant alleles at both the *arcC* and *glpF* loci; single nucleotide variants account for the remainder of the variation within this nodal cluster. The nodal subgroup around ST38 presents a different picture. Of the 10 allele changes occurring within the six STs in this subgroup, four can be attributed to recombination and the remaining six to mutation; thus both single site substitutions and recombination events contribute to the differences between STs within the subgroup. Overall, this assessment indicates the ratio of recombination to mutation to be about 8:32 in the 32 STs comprising the eBURST clonal network. Notably, there is only one example of allele sharing between STs in different nodal subgroups in the eBURST network: the variant allele *glpF-3* is shared between multiple STs in nodal subgroups ST15 and ST38. This allele is also shared with multiple STs outside the eBURST clonal network, validating its status as recombinant.

In contrast to the predominance of mutation over recombination in the eBURST clonal network, recombination events predominate in the seven STs containing HV alleles. Indeed, that these seven STs are comprised of mixtures of common core and HV alleles is indicative of recombination. Comparison of HV allele sequences at each of the five loci with HV alleles provides an estimated recombination to mutation ratio of 13:5. The phylogenetically supported branch containing ST18, ST25, and ST43 (Fig 2A) allows direct comparison at the ST sequence level and yields a recombination to mutation ratio of 8:2. The predominance of recombination to mutation among the HV alleles is indicative of the deeper ancestry of these alleles compared to those of the common core.

## Hypervariable alleles–remnants of a relict genotype?

The extreme sequence variation in the HV alleles relative to the common core alleles prompts the question of the origin of these alleles. To test the possibility the HV alleles are introgressions from other species, BLAST searches were done querying representative HV alleles against all genomes in the genus *Staphylococcu*s; no hits above 80% sequence identity were observed for any species other than *S. chromogenes*. Additionally, both the common core and HV allele sets are equidistant from the corresponding genes in *S. hyicus* reference sequences, consistent with expectation for common ancestry. An alternative hypothesis is that the HV alleles are remnants of a relict genotype. The large average pairwise SNP distances separating HV and common core alleles is indicative of an early time of divergence between the two classes of alleles (Table 3). The hypothesis that the HV alleles are remnants of a lineage older than the common core alleles is supported by the larger number of variant sites per allele for the HV alleles than for the common core alleles (6 vs. 1.04) and the higher average frequency of synonymous site variants in HV alleles than in the common core alleles (86.7% vs. 49.4%) [52]. Additional support for this hypothesis is the increased incidence of recombination relative to mutation in the HV alleles compared to the common core alleles; sequence variation due to recombination tends to accumulate over time.

The MLST profiles of the 7 outlier STs contain both common core and HV alleles, ranging from one to 5 HV alleles in an MLST profile. These mixed profiles are most likely to have arisen via recombination; mutational variation is not a plausible alternative. It is not possible to ascertain from the MLST data alone whether the mixtures are a result of introgression of non-HV alleles into an HV genome or the other way around. However, the apparent recent origin of the common core alleles and the possible relict origin of the HV alleles suggest the mixtures are relatively recent. A more detailed picture of the population history of *S. chromogenes* awaits further study using whole genome sequence data.

## Conclusions

The MLST scheme described in this paper provides a tool for the differentiation and identification of strains within *S. chromogenes*. With a power of discrimination between strain types exceeding 90% in geographically localized populations and greater than 95% overall, this MLST scheme has potential for use in epidemiological investigations of pathologies associated with this species and the ecological relationships between microbe and host. The geographic distribution of strain types indicated a high degree of genetic isolation between locales, posing a question of the historical and genetic factors accounting for this separation. Phylogenetic analysis of strain types identified by the scheme showed most to be contained within a single large and genetically diversified lineage which included strains arising from mutation driven clonal expansions and more varied strains generated by recombination events. The MLST analysis also revealed that some strain types were differentiated by having alleles with highly variable sequences at one or more of the loci in the 7-locus MLST scheme; these highly variable alleles were posited to be remnants of a relic genotype of *S. chromogenes*. These features of the population structure of this species provide a prospectus for future studies.

## Supporting information

**S1 Table. Isolate source data and classification.**
(XLSX)

**S2 Table. Genetic loci and primer sequences used in MLST scheme.**
(DOCX)

## Acknowledgments

We acknowledge the work of the staff The Vermont Integrative Genomics Resource DNA Facility, who completed the automated DNA sequencing.

## Author Contributions

**Conceptualization:** John W. Barlow, George Sensabaugh.

**Data curation:** Robert Mugabi, Gabriella Hetesy, John W. Barlow, George Sensabaugh.

**Formal analysis:** George Sensabaugh.

**Funding acquisition:** John W. Barlow.

**Investigation:** Robert Mugabi.

**Methodology:** Rebeca Huebner, Robert Mugabi, Gabriella Hetesy, John W. Barlow, George Sensabaugh.

**Project administration:** John W. Barlow.

**Resources:** Lawrence Fox, Sarne De Vliegher, Anneleen De Visscher, John W. Barlow.

**Supervision:** John W. Barlow.

**Writing – original draft:** George Sensabaugh.

**Writing – review & editing:** Robert Mugabi, Gabriella Hetesy, Lawrence Fox, Sarne De Vliegher, Anneleen De Visscher, John W. Barlow, George Sensabaugh.

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
