## [Decision Letter · Decision Letter 0]

30 Dec 2020

PONE-D-20-37271

Characterization of genetic diversity and population structure within Staphylococcus chromogenes by multilocus sequence Typing

PLOS ONE

Dear Dr. Barlow,

Thank you for submitting your manuscript to PLOS ONE. After careful consideration, we feel that it has merit but does not fully meet PLOS ONE’s publication criteria as it currently stands. Therefore, we invite you to submit a revised version of the manuscript that addresses each of the points raised during the review process.

We look forward to receiving your revised manuscript.

Kind regards,

D. Ashley Robinson, Ph.D.

Academic Editor

PLOS ONE

Journal Requirements:

Reviewers' comments:

Reviewer's Responses to Questions

**Comments to the Author**

1. Is the manuscript technically sound, and do the data support the conclusions?

Reviewer #1: Partly

Reviewer #2: Yes

2. Has the statistical analysis been performed appropriately and rigorously? 

Reviewer #1: Yes

Reviewer #2: I Don't Know

3. Have the authors made all data underlying the findings in their manuscript fully available?

Reviewer #1: Yes

Reviewer #2: Yes

4. Is the manuscript presented in an intelligible fashion and written in standard English?

Reviewer #1: Yes

Reviewer #2: Yes

5. Review Comments to the Author

Reviewer #1: This is a well written paper that focuses on the development of a mulilocus typing scheme for Staphylococcus chromogenes. It describes the genetic diversity and population structure of the organism, however, these data are derived from a relative small number of organisms obtained from only three geographic regions and only include a total of 46 sequence types whereas the pubmlst website currently includes 90 sequence types for this organism. If the goal is to study genetic diversity and population structure all of the available sequence types should be included. The usefulness of molecular typing of pathogenic bacteria is in their ability to identify strains associated with disease, virulence factors, host predilection, etc. For completeness the authors should associate each sample with the year of collection, host species, and the disease of their host, if any, with their sequence type.

Line 81 and others. The authors state that the seven loci used for their MLST are well separated around its genome. However, (line 135) this is based on a complete genome sequence of a different species, S. hyicus. The whole genome of S. chromogenes is available in GenBank (for example CP046028) and should be used for this purpose.

Line 100 Reference 31, presumably cited for identification of S. chromogenes using the tuf gene mentions this species only as being misidentified for another species with the API ID test and does not validate the use of this method for this organism. Validation of this method and a description of any other test used to identify the species of the samples should be included.

Table S1. Carbamate kinase is in contig 11 at SCHR_10075 not SCHR_ 09950.

Reviewer #2: The authors have developed the first MultiLocus Sequence Typing (MLST) scheme for typing Staphylococcus chromogenes to describe the genetic diversity of this species in relation to pathogenicity. Upon application of the scheme to 120 isolates collected from 3 locations (2 in the US and 1 from Europe, Belgium), the authors show (i) a common lineage including 39 of the 46 sequence types described, (2) a correlation between genotype and geographic location, and (3) a mainly mutational evolution of this dairy cattle mammary pathogen, even if recombination events generated outlier STs. The study is clearly presented and reasonable conclusions are drawn. Therefore, this report is interesting.

Here are the questions I feel the authors need to address before publication.

Major comments:

- Lines 27-28: it is stated that the aim of the study was to develop an MLST scheme in order to “facilitate study of strain variation as a factor of pathogenicity” but I do not see anywhere in the manuscript that pathogenicity was analyzed in relation to the MLST data even though the study included isolates from milk from healthy quarters and from cows with intramammary infections and others from commensal or environmental sources (lines 90-95).

- Lines 66-68: In addition to PFGE analyses, the authors should briefly mention that an MLVA scheme was also been developed for S. chromogenes typing (see Ruiz-Romero, DOI: 10.21307/pjm-2018-019)

- Lines 159-162: Please comment on these problems in the discussion section. The size of the seven amplicon is particularly large (Table S1). Will the scheme be easily applicable?

- The results and discussion section (> Line 187):

The results reported here are hardly discussed, if at all compared to the data in the literature. It would be interesting to confront the results of these phylogenetic analyses to that reported for other Staphylococcal species, for which an MLST was developed or for S. agnetis which is a closely related mammary pathogen. Are the evolutionary characteristics reported here highly specific to S. chromogenes?

The authors should also describe what MLST brings in relation to the phylogenetic relationships described for S. chromogenes by PFGE and MLVA.

The authors must also comment on the two different protocols used to prepare the cell pellets (see lines 108-119). Why did you make a different protocol between the two laboratories? what did it bring?

- Lines 87-88: Please give details about the period of collection of the isolates to better describe the geographic distribution of strain types.

- Lines 223-227: were the isolates from a single location (eg ST28, ST5) collected over several years? Were isolates with the same ST and from the same geographic place collected from several dairies? Is there a risk of analysing isolates with epidemiological links? Was the profile of antibiotic resistance of the isolates studied? What are the STs of the two pig isolates?

- Lines 231-232: how have the percentages of 90.2 and 93 been calculated?

- Lines 271-273: were the two ST44 isolates unambiguously identified as S. chromogenes? From what sample were they isolated?

Minor comments:

Line 195: use directly “SNP” as the abbreviation has already been introduced on line 164

Lines 199 to 200, Line 254-255: gene names (arC, dnaJ, etc.) must be written in italics

Line 237: use directly “SLV” as the abbreviation has already been introduced on line 174

Line 242: use directly “DLV” as the abbreviation has already been introduced on line 174

Lines 276-279: The sentence is particularly long. Thus, it is easy to read.

6. PLOS authors have the option to publish the peer review history of their article (what does this mean?). If published, this will include your full peer review and any attached files.

Reviewer #1: No

Reviewer #2: No

---

## [Author Response · Author response to Decision Letter 0]

13 Feb 2021

We thank the reviewers for their comments. We respond here to each point raised by the reviewers and the academic editor. We have also addressed the additional requirements addressed by the editorial staff, including file naming consistent with PLOS ONE’s style and removing a sentence in the discussion were we noted “data not shown.”

Reviewer #1: This is a well written paper that focuses on the development of a mulilocus typing scheme for Staphylococcus chromogenes. It describes the genetic diversity and population structure of the organism, however, these data are derived from a relative small number of organisms obtained from only three geographic regions and only include a total of 46 sequence types whereas the pubmlst website currently includes 90 sequence types for this organism. If the goal is to study genetic diversity and population structure all of the available sequence types should be included. 

AU: The primary objective of the paper is to report on the development of a MLST scheme that can be used to explore genetic diversity and population structure. Therefore, as a proof of concept for the possible utility, our study explored the genetic diversity and population structure for the 120 isolates shared between our laboratories during the development of the scheme. Most papers reporting a new scheme traditionally include a phylogenetic analysis to show the genetic relationships among and between STs for the isolates used. We believe the relationships we discovered when generating the phylogeny are an interesting result that paves the way for further analysis. Therefore we included the additional analysis on genetic diversity based on the sample population used for scheme development. We recognize that additional isolates have been added to the pubmlst database since we completed this scheme development work with the original 120 isolates. As the PubMLST database expands with continued submissions, researchers will be able to use these public data to explore the genetic diversity and population structure as the reviewer suggests.

The usefulness of molecular typing of pathogenic bacteria is in their ability to identify strains associated with disease, virulence factors, host predilection, etc. For completeness the authors should associate each sample with the year of collection, host species, and the disease of their host, if any, with their sequence type.

AU: These metadata are available at the PubMLST database. We have added a supplemental table that compiles these data for the 120 isolates used in this study (now S1 table). 

Line 81 and others. The authors state that the seven loci used for their MLST are well separated around its genome. However, (line 135) this is based on a complete genome sequence of a different species, S. hyicus. The whole genome of S. chromogenes is available in GenBank (for example CP046028) and should be used for this purpose.

AU: Thank you. When we originally developed the scheme, a complete genome of S. chromogenes was not available. We have updated the analysis and replaced the sentence previously at line 135. The new sentence (now at line 146) updates our work by locating the 7 candidate loci on the complete genome sequence of S. chromogenes strain 1401 [NZ_CP04602.1]. We have updated supporting information S1 table (now S2 table) incorporating this information.

Line 100 Reference 31, presumably cited for identification of S. chromogenes using the tuf gene mentions this species only as being misidentified for another species with the API ID test and does not validate the use of this method for this organism. Validation of this method and a description of any other test used to identify the species of the samples should be included.

AU: We have modified this sentence, adding a more recent reference (Capurro et al., 2009) instead of the Heikens reference. The Capurro reference includes a set of reference strains. We have also added more detail on the extent of species identification completed for the isolates included in this study. Therefore we are reasonably confident that the isolates in this study are in the S. chromogenes species group. To the best of our knowledge there are no prior comprehensive studies validating amplicon fragment sequence-based typing using tuf gene discriminating among staphylococcus species. In some studies ropB fragment typing has been used as the sole method and described as a “reference method” (e.g. Jenkins et al. 2019), and in this study we added a second method, tuf gene fragment sequence analysis, as added assurance. It is possible some isolates of the isolates in our sample are the closely related species S. hyicus or S. agnetis mis-identified as S. chromogenes. We believe this is highly unlikely, as these isolates have been variably typed with multiple methods. We have added a sentence to the results/discussion related to this issue.

Table S1. Carbamate kinase is in contig 11 at SCHR_10075 not SCHR_ 09950.

AU: now table S2, the contig naming has been updated in the table.

Reviewer #2: The authors have developed the first MultiLocus Sequence Typing (MLST) scheme for typing Staphylococcus chromogenes to describe the genetic diversity of this species in relation to pathogenicity. Upon application of the scheme to 120 isolates collected from 3 locations (2 in the US and 1 from Europe, Belgium), the authors show (i) a common lineage including 39 of the 46 sequence types described, (2) a correlation between genotype and geographic location, and (3) a mainly mutational evolution of this dairy cattle mammary pathogen, even if recombination events generated outlier STs. The study is clearly presented and reasonable conclusions are drawn. Therefore, this report is interesting.

Here are the questions I feel the authors need to address before publication.

Major comments:

- Lines 27-28: it is stated that the aim of the study was to develop an MLST scheme in order to “facilitate study of strain variation as a factor of pathogenicity” but I do not see anywhere in the manuscript that pathogenicity was analyzed in relation to the MLST data even though the study included isolates from milk from healthy quarters and from cows with intramammary infections and others from commensal or environmental sources (lines 90-95). 

AU: We have revise this sentence in the abstract to clarify the primary purpose of the manuscript, which is to report on the development of the scheme. We revised the sentence to communicate the concept of the potential utility of sequence-based typing schemes such as MLST. The study was not designed to explore potential relationships between sequence types (strains) and pathogenicity. We selected isolates from different geographic regions, sources and dates of collection only with the purpose of creating a potentially diverse sample set. We consider this a convenience sample, and there were no a priori considerations related to isolates source or time of collection. 

- Lines 66-68: In addition to PFGE analyses, the authors should briefly mention that an MLVA scheme was also been developed for S. chromogenes typing (see Ruiz-Romero, DOI: 10.21307/pjm-2018-019)

AU: We have added a sentence (now line 71 in the clean version) referencing this and other publications that have developed DNA fragment electrophoretic band pattern typing schemes (DNA band fingerprinting) for S. chromogenes. We have also added sentences describing the advantages of MLST or other DNA sequence-based systems compared to these fingerprinting methods.

- Lines 159-162: Please comment on these problems in the discussion section. The size of the seven amplicon is particularly large (Table S1). Will the scheme be easily applicable?

AU: These are not problems; these are standard practices for working with DNA sequences generated from Sanger sequencing (i.e. dye-terminator automated sequencing using capillary electrophoresis) in both directions to create a consensus sequence. Rather than considering this a problem, consider this a quality control step, such that agreement between the forward and reverse sequence is evaluated during alignment of the trace files (chromatograms). The size of the amplicons should not be an issue; sequencing amplicons of this length is standard practice for current chemistry systems. The size of the amplicons are purposely greater than the size of the informative sequence regions used for MLST so that the single strand overhangs and ambiguous reads at the ends can be readily trimmed without interfering with the informative sequence regions for the alleles. The ambiguous reads at the ends are typically about 20 base pairs where the automated capillary electrophoresis systems can not resolve difference between these small fragment sizes generate during dye-terminator sequencing. This scheme, like the many other MLST schemes will be easily applicable. In fact the PubMLST database is automated so that submissions of sequence reads are stripped of the non-informative ends and only the informative sequence fragments are used for comparison is evaluated. 

- The results and discussion section (> Line 187):

The results reported here are hardly discussed, if at all compared to the data in the literature. It would be interesting to confront the results of these phylogenetic analyses to that reported for other Staphylococcal species, for which an MLST was developed or for S. agnetis which is a closely related mammary pathogen. Are the evolutionary characteristics reported here highly specific to S. chromogenes?

AU: We have added to the results/discussion section. At line 265 we added two paragraphs describing how this MLST scheme can be used and comparing our results to the past DNA fingerprinting studies that have explored strain diversity among S. chromogenes isolates from different regions. 

The authors should also describe what MLST brings in relation to the phylogenetic relationships described for S. chromogenes by PFGE and MLVA.

AU: we have introduced (line 76) a description of what MLST brings in the introduction section, with an added reference (Maiden et al., 1998) that describes what sequence-based type schemes like MLST bring when compared to DNA fingerprinting systems including PFGE, MLVA and RAPD. We have also added a paragraph comparing the extent of diversity determined in prior studies using DNA fingerprinting methods and the challenges posed when comparing between studies that use these methods (line 267 – 272). 

The authors must also comment on the two different protocols used to prepare the cell pellets (see lines 108-119). Why did you make a different protocol between the two laboratories? what did it bring?

AU: Our two labs used pre-existing lab protocols for the preparation of isolate samples for DNA extraction. Both are standard protocols for primary culturing, colony collection, and colony growth. We simply reported this difference for transparency. We added clarification. 

- Lines 87-88: Please give details about the period of collection of the isolates to better describe the geographic distribution of strain types.

AU: We have added additional details to this sentence describing the periods when the isolates were collected. We have also added a supplemental table (S1) that collates the isolate meta-data from the PubMLST database.

- Lines 223-227: were the isolates from a single location (eg ST28, ST5) collected over several years? Were isolates with the same ST and from the same geographic place collected from several dairies? Is there a risk of analysing isolates with epidemiological links? Was the profile of antibiotic resistance of the isolates studied? What are the STs of the two pig isolates?

AU: We have added a supporting information table (now table S1) that collates the isolate meta-data from the PubMLST database. The study was not designed to test hypotheses related to the diversity of strains in time or geographic location (i.e. testing potential epidemiological associations with strain type). We selected isolates from different locations and times as readily available from the collections of the laboratories in Vermont, Washington and Belgium with the primary objective of increasing strain diversity. Isolates were not selected based on source, or disease status, or host species in a balanced retrospective cross-sectional study. Yes, there is a risk of analyzing isolates with epidemiological associations. Isolates are clustered in a hierarchy of country, state, region, farm, source. In fact, in the sample set assembled for scheme development a number of factors are likely confounded (e.g. year of isolate collection and location) so tests of association are not advisable. We have added references to prior typing studies exploring epidemiological associations. We do not report the profile of antimicrobial susceptibility among these isolates as that was not an objective of this study. The STs of the two pig isolates are 22 and 23; they are singleton outliers in the eBurst analysis defining the clonal complexes within these samples.

- Lines 231-232: how have the percentages of 90.2 and 93 been calculated?

AU: The discriminatory power is based on haplotype diversity. We revised the sentence clarifying this approach in the methods (now line 189).

- Lines 271-273: were the two ST44 isolates unambiguously identified as S. chromogenes? From what sample were they isolated?

AU: These isolates were from milk samples from Belgian cattle with intramammary infections. The species was determined by multiple methods, including transfer RNA intergenic spacer PCR or 16S rRNA gene fragment sequencing by our Belgian colleagues, and ropB and tuf gene fragment sequencing in the Vermont laboratory. We have added more details on species identification in the methods section (now lines 112-119). The difference between ST44 and its neighbors in the ST38 cluster is due solely to nucleotide sequence variation at the arcC and glpF loci. The other 5 alleles in the ST44 MLST profile are present in numerous other SCHR STs. See ST44 entry in table 2.

Minor comments:

Line 195: use directly “SNP” as the abbreviation has already been introduced on line 164

AU: corrected

Lines 199 to 200, Line 254-255: gene names (arC, dnaJ, etc.) must be written in italics

AU: corrected

Line 237: use directly “SLV” as the abbreviation has already been introduced on line 174

AU: corrected

Line 242: use directly “DLV” as the abbreviation has already been introduced on line 174

AU: corrected

Lines 276-279: The sentence is particularly long. Thus, it is easy to read.

AU: corrected

---

## [Editor Report · Decision Letter 1]

24 Feb 2021

Characterization of genetic diversity and population structure within Staphylococcus chromogenes by multilocus sequence typing

PONE-D-20-37271R1

Dear Dr. Barlow,

We’re pleased to inform you that your manuscript has been judged scientifically suitable for publication and will be formally accepted for publication once it meets all outstanding technical requirements.

Kind regards,

D. Ashley Robinson, Ph.D.

Academic Editor

PLOS ONE
---

## [Editor Report · Acceptance letter]

3 Mar 2021

PONE-D-20-37271R1 

Characterization of genetic diversity and population structure within *Staphylococcus chromogenes* by multilocus sequence typing 

Dear Dr. Barlow:

I'm pleased to inform you that your manuscript has been deemed suitable for publication in PLOS ONE. Congratulations! Your manuscript is now with our production department. 

Kind regards, 

on behalf of

Dr. D. Ashley Robinson 

Academic Editor

PLOS ONE